# Time-dependent study of disordered models
# with infinite projected entangled pair states

**Claudius Hubig[1,2⋆] and J. Ignacio Cirac[1,2]**

**1** Max-Planck-Institut für Quantenoptik, Hans-Kopfermann-Str. 1, 85748 Garching, Germany
**2** Munich Center for Quantum Science and Technology (MCQST),
Schellingstr. 4, 80799 München, Germany

⋆ claudius.hubig@mpq.mpg.de

## Abstract

Infinite projected entangled pair states (iPEPS), the tensor network ansatz for two-dimensional systems in the thermodynamic limit, already provide excellent results on ground-state quantities using either imaginary-time evolution or variational optimisation. Here, we show (i) the feasibility of real-time evolution in iPEPS to simulate the dynamics of an infinite system after a global quench and (ii) the application of disorder-averaging to obtain translationally invariant systems in the presence of disorder. To illustrate the approach, we study the short-time dynamics of the square lattice Heisenberg model in the presence of a bi-valued disorder field.



# 1  Introduction

Tensor networks are a powerful technique for the study of strongly-correlated many-body problems. In particular, ground states of local Hamiltonians are believed to be efficiently approximated by tensor network states (TNS). In one dimension, the density matrix renormalisation group (DMRG, [1]) and the associated tensor network states called matrix-product states (MPS) are the method of choice for the evaluation of ground-state observables [2]. In two dimensions, despite the worse scaling of computational effort, infinite projected entangled pair states (iPEPS, [3, 4]), the two-dimensional analogue of MPS, are already producing excellent results, e.g. Refs. [5–7].

The study of dynamics using TNS is more difficult as the growth of the entanglement entropy during a real-time evolution is typically unfavourable. This usually limits the simulation of dynamics to relatively short times. However, higher computational resources and new time-evolution methods [8] have made the study of dynamics in one dimension commonplace with many interesting applications, such as the calculation of excitation spectra, the use as solvers in dynamical mean-field theory and the real-time simulation of local and global quantum quenches.

In two dimensions, the situation is more difficult. Early studies of time-evolution of a finite PEPS [9] exist, but the method has not seen wide-spread application. One reason is the lack of gauge fixing in tensor networks with loops, which makes calculations more costly and unstable. Additionally, most of the development was focused on the infinite iPEPS ansatz which imposes translational invariance and hence restricts the problems one may study.

In this paper we show how to study the dynamics of certain disordered systems by simulating the time-evolution of an iPEPS. To make the disordered problem translationally invariant, we employ the ancilla trick introduced in Ref. [10]. This trick introduces ancilla spins on every site of the lattice. Coupling the ancilla spins to the physical degrees of freedom and initialising the ancillas in a superposition of spin-up and spin-down states leads to a local superposition of discrete disorder configurations. The tensor product of multiple sites results in an implicit averaging over all possible discrete disorder realisations which is translationally invariant and can be represented using iPEPS. To then simulate the dynamics of this system, we use real-time evolution instead of imaginary-time evolution while paying close attention to the numerical stability of the ansatz and the errors incurred.

To illustrate the method, we simulate the dynamics of the XXX Heisenberg model with a random bi-valued magnetic field; a similar setup was already studied in one dimension using similar techniques [11]. While we expect thermalisation to occur at sufficiently long times as clusters with the same disorder field eventually couple, here we aim at short time scales because, just as in one-dimensional MPS simulations, we are limited in the times we can obtain due to the entanglement growth in the system. However, even short-time simulations are of interest when comparing to recent experiments [12] which likewise study the real-time dynamics of two-dimensional disordered systems. Our intent is then to find qualitative effects of disorder on such short time scales.

To this aim, the rest of the paper is structured as follows: Section 2 revisits the ancilla trick to incorporate the disorder averaging. Section 3 introduces our physical test model and discusses the expected behaviour at infinite disorder strength. Section 4 describes the steps taken to obtain reliable real-time data within the iPEPS framework. Finally, Section 5 discusses our results obtained from the disordered two-dimensional Heisenberg model before drawing conclusions in Section 6.

## 2 Averaging via auxiliary spins

We use infinite projected entangled pair states [4] to study the two-dimensional system in the thermodynamic limit. The tensor network ansatz is based on a finite unit cell of tensors repeated infinitely whose (ideal) contraction represents the infinite quantum state of (e.g.) physical $S = 1/2$ spins. This ansatz requires some kind of translational invariance of the state e.g. under translation of 2 sites to the left. Such translational invariance is incompatible with randomly disordered systems. To avoid this problem, we restrict the disorder to be discrete and bi-valued, that is, for any disorder configuration $\mathcal{A}$, the function $z_{i;\mathcal{A}}$ assigns each lattice site a value $\pm 1/2$. We then follow Ref. [10] and introduce ancilla $S = 1/2$ spins on each lattice site $i$, doubling the local space (with $p$ and $a$ denoting physical and ancilla space respectively):

$$|\sigma_i\rangle \rightarrow |\sigma_i\rangle_p \otimes |\sigma_i\rangle_a \ . \tag{1}$$

We then initialise all ancilla spins in the $|+\rangle$ state

$$|+_i\rangle_a = \frac{1}{\sqrt{2}} \left( |\uparrow_i\rangle_a + |\downarrow_i\rangle_a \right) \ \forall i, \tag{2}$$

resulting in an initial ancilla state of $N$ sites

$$|+\rangle_a = \left( |+_i\rangle_a \right)^{\otimes N} = \frac{1}{\sqrt{2^N}} \sum_{\sigma_i = \uparrow, \downarrow} |\sigma_1 \sigma_2 \ldots \sigma_N\rangle_a \equiv \frac{1}{\sqrt{2^N}} \sum_{\substack{\text{disorder} \\ \text{config. } \mathcal{A}}} |\mathcal{A}\rangle_a \ . \tag{3}$$

For example, the tensor product of e.g. 3 ancilla spins is the state

$$|+_1\rangle_a \otimes |+_2\rangle_a \otimes |+_3\rangle_a = \frac{1}{\sqrt{8}} \left( |\uparrow_1\rangle_a + |\downarrow_1\rangle_a \right) \otimes \left( |\uparrow_2\rangle_a + |\downarrow_2\rangle_a \right) \otimes \left( |\uparrow_3\rangle_a + |\downarrow_3\rangle_a \right) \tag{4}$$

$$= \frac{1}{\sqrt{8}} \left( |\uparrow_1 \uparrow_2 \uparrow_3\rangle_a + |\uparrow_1 \uparrow_2 \downarrow_3\rangle_a + |\uparrow_1 \downarrow_2 \uparrow_3\rangle_a + |\uparrow_1 \downarrow_2 \downarrow_3\rangle_a \tag{5}$$

$$+ |\downarrow_1 \uparrow_2 \uparrow_3\rangle_a + |\downarrow_1 \uparrow_2 \downarrow_3\rangle_a + |\downarrow_1 \downarrow_2 \uparrow_3\rangle_a + |\downarrow_1 \downarrow_2 \downarrow_3\rangle_a \right), \tag{6}$$

i.e. an equal superposition of all possible $\uparrow, \downarrow$ configurations. We now consider the physical Hamiltonian for a specific disorder realisation $\mathcal{A}$

$$\hat{H}_{\mathcal{A};p} = \sum_{\langle i,j \rangle} \hat{s}_{i;p} \cdot \hat{s}_{j;p} + h \sum_i \hat{s}^z_{i;p} z_{i;\mathcal{A}} \ , \tag{7}$$

where $\hat{s}_{i;p}$ are physical spin operators, $\hat{s}^z$ denotes the $z$-component of the spin operator only, $\langle i,j \rangle$ denotes nearest-neighbour sites $i = x, y$ and $j = x, y \pm 1$ or $j = x \pm 1, y$, the effective disorder field has a strength $h \geq 0$ and the function $z_{i;\mathcal{A}}$ assigns each lattice site a value $\pm 1/2$. Now the time-dependent disorder-averaged expectation value of $\hat{O}_p$ starting from some initial

physical state $|\psi\rangle$ is

$$\langle\langle \hat{O}_p(t)\rangle\rangle \equiv \frac{1}{2^N} \sum_{\substack{\text{disorder} \\ \text{config. } \mathcal{A}}} {}_p\langle\psi|\hat{O}(t)|\psi\rangle_p \tag{8}$$

$$= \frac{1}{2^N} \sum_{\substack{\text{disorder} \\ \text{config. } \mathcal{A}}} {}_p\langle\psi|e^{it\hat{H}_{\mathcal{A};p}}\hat{O}e^{-it\hat{H}_{\mathcal{A};p}}|\psi\rangle_p \tag{9}$$

$$= \frac{1}{2^N} \sum_{\substack{\text{disorder} \\ \text{config. } \mathcal{A}}} \left({}_a\langle\mathcal{A}|\otimes{}_p\langle\psi|\right)e^{it\hat{H}_{\mathcal{A};p}}\hat{O}e^{-it\hat{H}_{\mathcal{A};p}}\left(|\psi\rangle_p\otimes|\mathcal{A}\rangle_a\right) \tag{10}$$

$$= \frac{1}{2^N} \sum_{\substack{\text{disorder} \\ \text{config. } \mathcal{A}}} \left({}_a\langle\mathcal{A}|\otimes{}_p\langle\psi|\right)e^{it\hat{H}}\hat{O}e^{-it\hat{H}}\left(|\psi\rangle_p\otimes|\mathcal{A}\rangle_a\right) \tag{11}$$

$$= \left({}_a\langle+|\otimes{}_p\langle\psi|\right)e^{it\hat{H}}\hat{O}e^{-it\hat{H}}\left(|\psi\rangle_p\otimes|+\rangle_a\right), \tag{12}$$

with the effective Hamiltonian

$$\hat{H} = \sum_{\langle i,j\rangle}\hat{s}_{i;p}\cdot\hat{s}_{j;p} + h\sum_i \hat{s}_{i;p}^z\hat{s}_{i;a}^z \tag{13}$$

and $\hat{s}_{i;a}$ the auxiliary spin operator on site $i$.

## 3 The Heisenberg model with discrete disorder

The Hamiltonian Eq. (13) and the initial state $|\psi\rangle_a$ of the ancilla spins is translationally invariant. We initialise the physical spins in a Néel state (i.e., checkerboard pattern), which can be described by a $2\times 2$ unit cell.

In this situation, at $h = \infty$, dynamics are constrained to clusters of the same disorder configuration. Those clusters are randomly sized and shaped and will support no dynamics ($1\times 1$ "cluster"), some oscillations (small clusters) or essentially thermodynamic-limit like behaviour (for very large clusters). In any specific disorder realisation, any particular site has probability $p = 1/2$ of belonging to the ancilla-up or ancilla-down type. This probability is below the critical percolation probability $p_c \approx 0.592$ [13, 14] of the square lattice site percolation problem [15]. As a result, there is no cluster of infinite size and the probability of a given cluster occurring decays exponentially with its size. Because odd cluster sizes are more common than even cluster sizes (1 is more common than 2, 3 is more common than 4 etc.), the sign of the average magnetisation of a given cluster even at long times will coincide with the initial magnetisation of the chosen site.

A first extension of the present study would be $S = 1$ auxiliary ancilla spins. There, too, cluster probability decays exponentially in cluster size, but this decay is much faster. One hence expects stronger localisation effects and less pronounced oscillatory dynamics. In the limit of infinite local ancilla spins, each cluster will have size 1 and the system will hence be completely frozen (still at $h = \infty$), corresponding to a standard model with infinitely strong continuous disorder.

Going away from $h = \infty$ to smaller disorder strengths then leads to some coupling between clusters. Since clusters with equal configurations of ancilla spins can couple through potential barriers of opposite-orientation ancilla spins at times linear in the disorder strength, we expect eventual equilibration for discrete disorder configurations as generated by finite ancilla spins. However, even if equilibration occurs eventually, we expect to see an effect in the short-term dynamics after a global quench.

# 4 Real-time evolution in iPEPS

Real-time evolution of an iPEPS is "technically straightforward" [9, 16], it suffices to replace the standard prefactor $\beta$ used during imaginary-time evolution to find the ground state by a real time step $it$. In practice, the quick growth of entanglement and instability of the iPEPS is problematic. Here, we are using a second-order Trotter-Suzuki decomposition [17] of the Hamiltonian Eq. (13). There are four bonds inside the $2 \times 2$ unit cell and four bonds linking each unit cell to its right and upper neighbour. With the symmetrisation due to the second-order decomposition, we hence have to apply 16 individual gates to perform a single time step. The innermost two gates can be combined to reduce the gate count by one. After each individual gate application, we then use the full update [4] to recompute the environment for maximal accuracy. During the simulation, we use the U(1)-$S^z$ symmetry of the system in the physical sector [18] which allows us to take the bond dimension up to $D = 9$. Each site of the iPEPS has local dimension 4 as we combine physical and ancilla spins into individual sites. The initial Néel state is represented by an iPEPS with bond dimension $D = 1$.

Contrary to MPS, iPEPS cannot be gauged [16, 19, 20] perfectly. Using a bond dimension $D$ in an iPEPS to represent a state which would only require a much smaller bond dimension $D' \ll D$ leads to serious problems with numerical stability in both the corner-matrix renormalisation group procedure as well as the next full update due to a badly-conditioned norm tensor. As entanglement grows, we need to increase the bond dimension during some time steps from $D$ to $D + 1$ and hence have to balance multiple objectives: First, small time steps lead to a small Trotter error, which is desired. Second, large time steps allow increasing the bond dimension in a very stable manner, as the change induced by the step is large and hence additional entanglement between the two involved sites is created. Third, even in a series of large time steps, we cannot grow the bond dimension at every step and hence have to select the optimal step at which numerical stability is ensured and the loss of information due to too strong a truncation is minimised. Hence increasing the bond dimension too quickly leads to problems with numerical stability, increasing it too late constrains the dynamics to a lowly-entangled subspace. In an imaginary-time evolution to find the ground state, one can typically ignore this problem and accept some numerical instability for a few steps after increasing $D$. Here, we do not have that luxury and need to be precise and stable during all steps.

Overall, there are three free parameters in our calculation: first, the step size $\delta t$ used in the Trotter decomposition. We select $\delta t = 0.1, 0.05, 0.025$ and $0.01$ and check that we obtain relatively converged results between those. Second, we measure the accumulated cost of truncation during a full time step as the sum of the norms of differences between truncated and untruncated states at each individual gate application and, once this cost crosses a threshold $\varepsilon$, increase the bond dimension $D$ by 1. We select $\varepsilon = 10^{-1,-2,-3,-4}$ and $5 \cdot 10^{-2}$. Finally, the physical parameter $h$ couples the ancilla spins to the physical spins. We select $h = 0, 1, 4, 16, 64, 256$. The environment bond dimension $\chi$ is set to $10(D + 1)$ throughout the evolution, testing suggests that this is sufficient.

## 4.1 Converging results in $\delta t$ and $\varepsilon$

For fixed $h$, we then compare the results at different $\varepsilon$ and $\delta t$. The evaluated expectation values are $\langle \hat{H} \rangle$ and $\langle \hat{s}^z_{0,0;p} \rangle$ on the first unit cell site. The spins on the other three unit cell sites behave as expected as $\pm \langle \hat{s}^z_{0,0;p} \rangle$ to very good accuracy.

Some of the calculations are still unstable (e.g. $\varepsilon = 10^{-4}$ tends to be problematic) leading to a very abrupt and large error in the energy; this problem is more pronounced at larger values of $h$ and smaller step sizes $\delta$. The instability is due to an "insufficient" growth of entanglement in the iPEPS which does not fill up the allowed bond dimension $D$. This situation can be

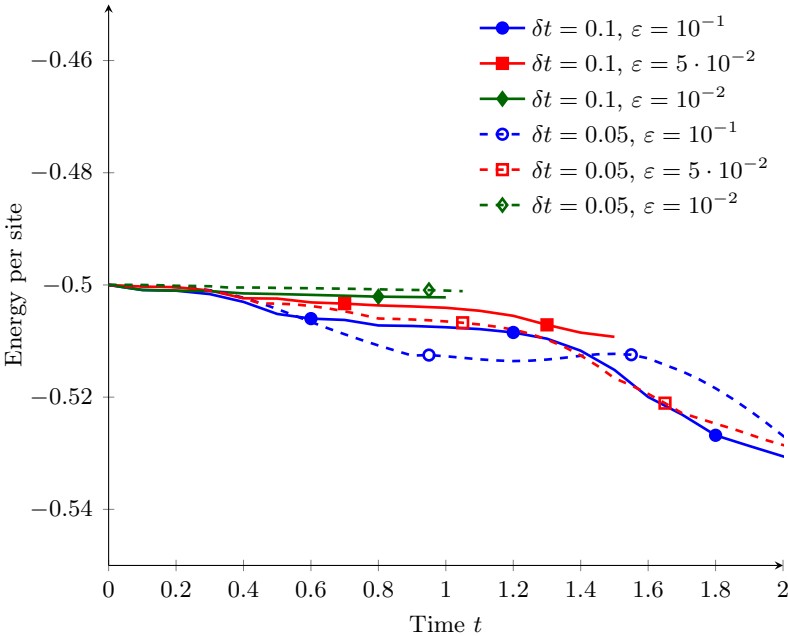

Figure 1: Energy per site over time at $h = 4$ as obtained from some stable example configurations for $\delta t$ and $\varepsilon$. For the later plots Figs. 3 and 4, we have selected $\delta t = 0.1$, $\varepsilon = 0.05$ (solid red squares) to represent $h = 4$ as the error is sufficiently small.

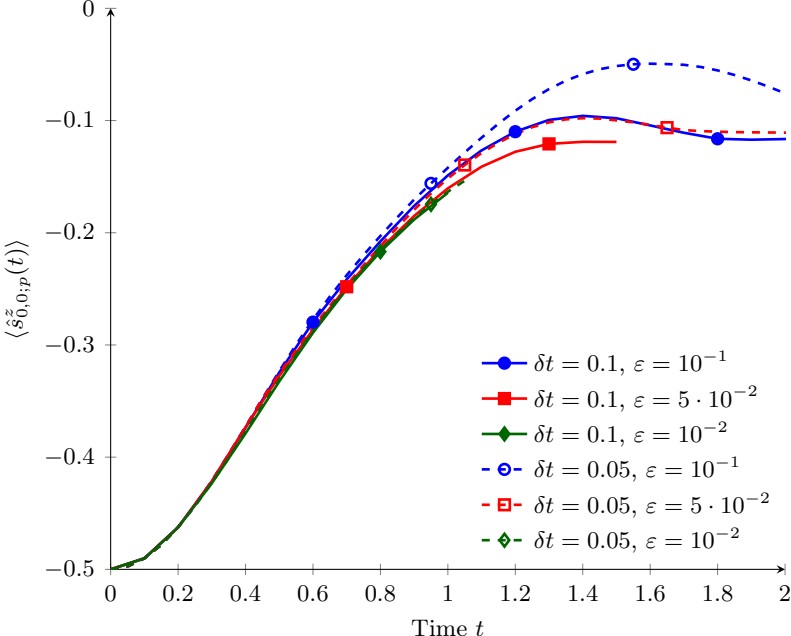

Figure 2: Expectation value of $\langle \hat{s}^z_{0,0;p}(t) \rangle$ at $h = 4$ for the calculations also shown in Fig. 1. The selected setting $\delta t = 0.1$, $\varepsilon = 0.05$ is very close to the higher-precision calculations with $\varepsilon = 10^{-2}$, though those exhaust the available bond dimension earlier.

diagnosed reliably and the associated calculations are discarded.

Furthermore, if $\varepsilon$ is too large and the time step $\delta t$ is too small, a large error is made at every step which over time leads to no entanglement growth (even at $h = 0$!) and instead to wrong oscillatory dynamics over a limited range. Diagnosing this problem based only on the energy and spin expectation values is very difficult, as the energy is conserved relatively very well and the spin expectation values form a smooth curve. We instead focus on the growth of the bond dimension $D$. If during the calculation the bond dimension does not grow to the maximal value $D = 9$, we assume that the truncation error made is too large and the calculation is discarded.

The above two criteria discard a number of calculations, especially at small step sizes and very large or very small error threshold. From the rest, we obtain fairly well-converged data typically at $\delta t \leq 0.05$ and $\varepsilon = 10^{-2}$. Figs. 1 and 2 show exemplary data curves at $h = 4$ once the unstable and wrong results were removed. We obtain good convergence of the local observable $\langle \hat{s}^z_{0,0;p}(t) \rangle$ for a range of time steps $\delta t = 0.1, 0.05$ and error thresholds $\varepsilon = 10^{-1}, 5 \cdot 10^{-2}, 10^{-2}$ and $10^{-3}$ (not shown). One calculation at $\varepsilon = 10^{-1}$ and $\delta = 0.05$ produces very different dynamics from the others likely due to the large truncation error. Decreasing the truncation error at fixed time step size or increasing the time step size at fixed truncation error per step leads to a number of very similar results.

## 5 Disorder-averaged dynamics

Using the procedure above, we obtain one converged curve $\langle \hat{s}^z_{0,0;p}(t) \rangle$ per disorder coupling strength $h$. Depending on the entanglement growth in the system and the threshold $\varepsilon$ selected, the calculations exceed the available resources at different times $t$. As a result, not all calculations reach our maximal target time $t = 2$. Fig. 3 shows the energy per site in these calculations. We observe the energy to be conserved relatively well with an error comparable in magnitude to the cost function measured during each iPEPS full update, the relative error in energy per site is never higher than a few per cent even after many full-update steps. We do not take any particular steps to enforce energy conservation during the update. The error in energy can hence be – to some extend – considered an error measure for our calculation itself. As a side effect of the Trotterisation, in particular the energy conservation is affected by the Trotter error. It is then understandable that at larger disorder strengths $h$, the commutator between the local field terms $h \hat{s}^z_{i;p} \hat{s}^z_{i;a}$ and bond terms $\hat{s}^x_{i;p} \hat{s}^x_{j;p}$ necessarily becomes larger and we observe worse energy conservation.

In Fig. 4 we analyse the time-dependent expectation value $\langle \hat{s}^z_{0,0;p}(t) \rangle$. Very weak disorder $h = 1$ leads to essentially the same dynamics as the case without disorder ($h = 0$). At $h = 4$, the initial dynamics ($t \leq 0.5$) are still comparable to those of the clean system and only later times ($t \approx 1$) show an effect due to the disorder. Increasing the disorder strength further we find distinctly different dynamics at $h \geq 16$. These differences set on very early ($t \approx 0.1$) and lead to noticeably weaker oscillations, with the first peak at $t \approx 1$ at $\langle \hat{s}^z_{0,0;p}(1) \rangle \approx -0.3$. Of course, we would expect that stronger disorder leads to monotonically increasing differences in the dynamics, which does not appear to be the case in Fig. 4. The obtained data range at $h = [16, 64, 256]$ may hence also serve as an approximate error measure for our calculation. Nevertheless, a drastic difference to data at $h \leq 4$ is visible, both in the very precise shortest-time frame $t \leq 0.5$ and at longer times $t \geq 1$.

By comparing with time-dependent MPS calculations [21] on 4×4 and 6×6 tori in the zero-disorder case, we find finite-size effects at time $t > 0.5$. Comparable or smaller system sizes are available to exact diagonalisation studies while numerical linked cluster calculations on similar systems [22] likewise encounter finite-size effects around $t < 1$. Our method, extending the

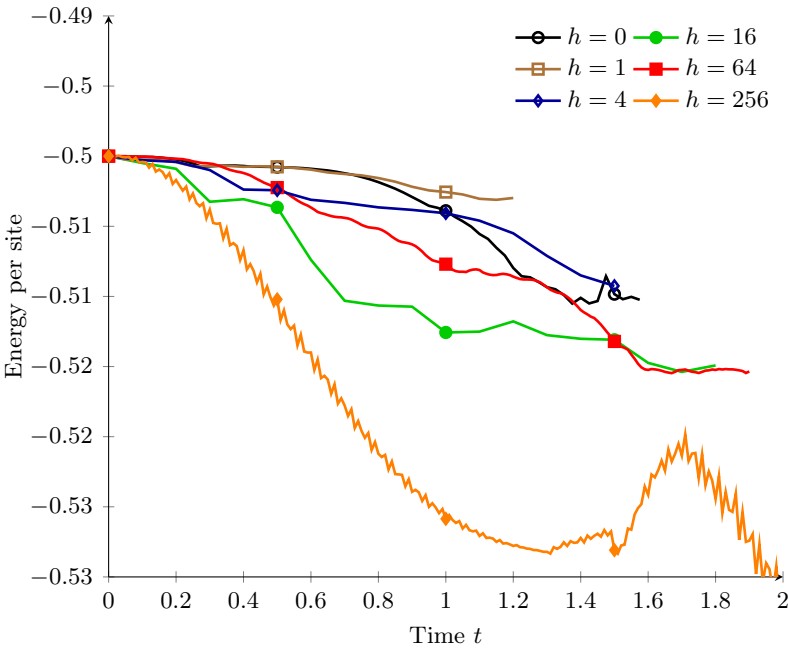

Figure 3: Energy per site over time at various disorder strengths. In all calculations, the error in energy is relatively small and typically smaller than the accumulated cost of truncation.

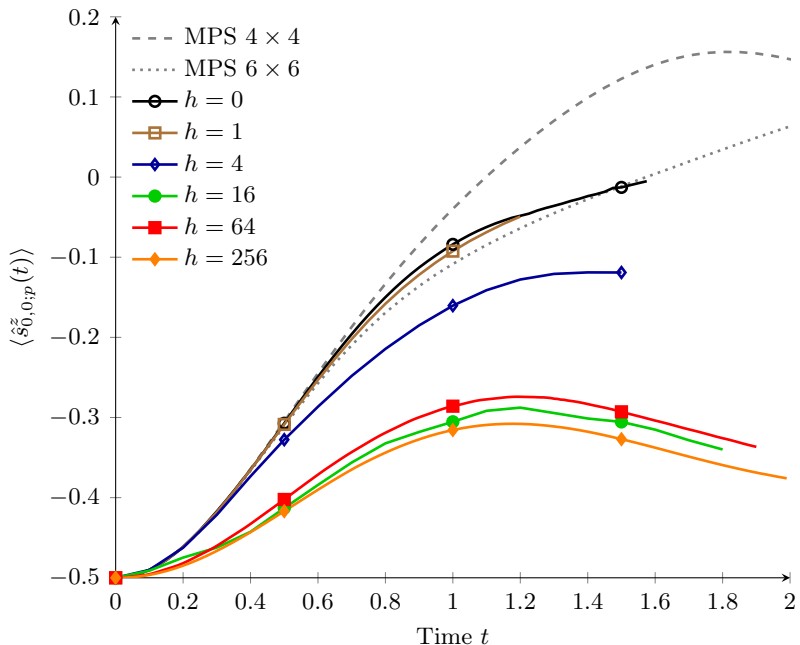

Figure 4: Expectation value of $\langle \hat{s}^z_{0,0;p}(t) \rangle$ at various disorder strengths. A noticeable slow-down of dynamics is observed at increased disorder strengths $h \geq 16$; MPS calculations at $h = 0$ show finite-size effects at $t < 1$.

time frame to at least $t \approx 1.5$, is hence certainly an improvement over alternative MPS-based approaches. Additionally, future improvements to increase the stability of the calculation and further increase the bond dimension should help in obtaining longer times without having to worry about finite-size effects.

## 6  Conclusion

Presently, the iPEPS formalism allows for the simulation of the real-time evolution of quantum states free from boundary effects. While limited in the obtainable times due to entanglement growth in the present system, this limit is still beyond the onset of finite-size effects in comparable matrix-product state, exact diagonalisation or numerical linked cluster approximation calculations. Using the formalism, we were able to study a discrete-disorder-averaged two-dimensional Heisenberg model, where we have found a slowdown of dynamics at disorder coupling strengths $h \geq 16$. This slowdown is markedly different from the dynamics encountered in the zero-disorder case. The generalisation to multi-valued disorder configurations will be the topic of future work.

The two primary challenges at the moment are the stability of the full update, which limits the obtainable precision by requiring a certain minimal cost of truncation at each step and the relatively large Trotter error induced by the second-order decomposition of the time evolution exponential. The stability of the full update strongly depends on the choice of the corner transfer matrix renormalisation growth and we are positive that improved methods – potentially based on better gauging [23] or a canonical-like form for PEPS [24] – will be introduced. Likewise, the large Trotter error encountered here could be handled either by improved Trotter decompositions [25], a hybrid approach to evolve more than two sites exactly [26] or by adapting the variational ground-state search [7,27] to work as a two-dimensional time-dependent variational principle.

## Acknowledgements

During the course of this project, Ref. [28] similarly applied real-time evolution of iPEPS to the transverse field Ising model.

**Funding information**     The authors acknowledge funding through ERC Grant QUENOCOBA, ERC-2016-ADG (Grant no. 742102) and by the Deutsche Forschungsgemeinschaft (DFG, German Research Foundation) under Germany's Excellence Strategy – EXC-2111 – 390814868.

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
