# Peer review of "Time-dependent study of disordered models with infinite projected entangled pair states"

_SciPost Physics, doi:SciPost Phys. 6, 031 (2019)_

## Round 1 · Referee Report · Anonymous · 2019-1-7

Strengths
1. The paper addresses a challenging and highly relevant subject
2. This is the one of the pioneering works on real-time evolution and disordered systems in 2D with iPEPS which will serve also as a benchmark for future works.
3. The authors provide a useful discussion of the convergence behavior and stability issues in their simulations
Weaknesses
1. The error due to the truncation in Fig. 4 is not clear (see report)
2. The conclusion section is somewhat minimalistic and could be improved (see report)
3. There are a few minor points in the paper which can be made clearer (see report)
Report
In this paper the authors show how infinite projected entangled-pair states (iPEPS) can be applied to perform a real-time evolution after a global quench of a disordered system, using the full update and the recently introduced ancilla approach. As an example they study the XXX Heisenberg model with a random bi-valued magnetic field.
Real-time evolution in 2D is very challenging and in this work the authors demonstrate how far one can currently go with iPEPS in this context which I find an interesting and valuable contribution in the field of tensor networks. This work also provides a nice application and 2D extension of the ancilla approach which was recently introduced for matrix product states to study disordered systems. The authors provide a useful discussion of the convergence behavior and stability issues in their simulations.
For all these reasons I can recommend publication of this article in Scipost after the authors have addressed the points mentioned below.
Comments and questions:
1) Can the authors define the "accumulated cost of truncation" more precisely? Is it the sum of all norm differences between the truncated and untruncated wave function in a full time step?
2) Caption Figure 3: Which error do the authors mean by "All results have relatively small errors". The error on the conservation of the energy or another one? This could be written in a clearer way.
3) In Fig. 4 it would be interesting to see the behavior for more values at intermediate disorder strength, i.e. in between h=4 and h=16, to see how quickly the large-h dynamics is reached.
4) Fig. 4: it would be good to add h=0 to the legend for the MPS data, and/or in the figure caption.
5) What are the parameters of epsilon and time step used in Sec. 5 (Figures 3 and 4)? In the end the parameter of epsilon determines also the accuracy of the calculation and it would be useful for each value of h to show data for at least two different epsilon values to have an idea of the truncation effects. It seems difficult to know how large these truncation errors really are (e.g. compared to the finite size error in the 6x6 MPS); ideally one would want to extrapolate in epsilon - but I guess this is tricky.
6) The conclusion section is somewhat minimalistic and the first sentence seems to me an overstatement, since the errors are only controlled up to a certain extent; e.g. with the current data it would be difficult to put a reliable error bar on the local observable at time t=1.5. It would be good to revise the conclusion part, to provide a more detailed summary of their work and findings, and to add a brief outlook.
7) In the author's approach the bond dimension is increased by 1 whenever the truncation is above a certain threshold epsilon. Is an increase by 1 always enough in practice? What would happen if the authors run the simulations at Dmax=9 from the start, rather than gradually increasing D?
Requested changes
See points 1)-6) in the report
Author: Claudius Hubig on 2019-02-15 [id 441]
(in reply to Report 1 on 2019-01-07)
We would like to thank the referee for their work and helpful comments and have replied in detail in-line below.
1) Can the authors define the "accumulated cost of truncation" more precisely? Is it the sum of all norm differences between the truncated and untruncated wave function in a full time step?
Yes, we have clarified this.
2) Caption Figure 3: Which error do the authors mean by "All results have relatively small errors". The error on the conservation of the energy or another one? This could be written in a clearer way.
We have clarified the caption, we indeed intend to refer to the error in the energy.
3) In Fig. 4 it would be interesting to see the behavior for more values at intermediate disorder strength, i.e. in between h=4 and h=16, to see how quickly the large-h dynamics is reached.
We have completed some additional calculations at h = 6, 8, 10, 12, 14. This data suggests major changes between h=4, 6 and 8. 8 and 10 already provide similar results on the timescaled we can access. However, given the still relatively large errors of these results, we feel both more comfortable pointing out the drastic difference between h=4 and 16 and also consider Fig. 4 with substantially more data lines to be too confusing.
4) Fig. 4: it would be good to add h=0 to the legend for the MPS data, and/or in the figure caption.
Agreed, we have amended the caption accordingly.
5) What are the parameters of epsilon and time step used in Sec. 5 (Figures 3 and 4)? In the end the parameter of epsilon determines also the accuracy of the calculation and it would be useful for each value of h to show data for at least two different epsilon values to have an idea of the truncation effects. It seems difficult to know how large these truncation errors really are (e.g. compared to the finite size error in the 6x6 MPS); ideally one would want to extrapolate in epsilon - but I guess this is tricky.
This is indeed problematic, as contrary to MPS calculations which give successively better results at smaller truncation thresholds, our iPEPS implementation becomes unstable below a certain threshold. As such, we only have a "window", not a half-line of successively better data in which to converge our results. The values of epsilon and time step delta chosen for Fig. 3 and 4 where comparable to those selected in Fig. 2, i.e.~O(10^-2) for epsilon and O(10^-1) for the time step. The effects due to different epsilons and delta are also very comparable between different values of the disorder strength h; we hence believe Fig. 1 and 2 to be sufficient examples.
6) The conclusion section is somewhat minimalistic and the first sentence seems to me an overstatement, since the errors are only controlled up to a certain extent; e.g. with the current data it would be difficult to put a reliable error bar on the local observable at time t=1.5. It would be good to revise the conclusion part, to provide a more detailed summary of their work and findings, and to add a brief outlook.
We have revised the Conclusions and in particular extended the outlook to include possible avenues for future improvements, work to be done and ongoing efforts moving us into this direction.
7) In the author's approach the bond dimension is increased by 1 whenever the truncation is above a certain threshold epsilon. Is an increase by 1 always enough in practice? What would happen if the authors run the simulations at Dmax=9 from the start, rather than gradually increasing D?
Especially at large disorder strengths, the instability problems due to imperfect gauging of the iPEPS are very severe. Any calculation would hence immediately fail. At small or no disorder, this problem is less severe and e.g. Fig. 2 suggests that the overall effect of bond dimension at short times (where it is not maximal at the moment) is very small.
Author: Claudius Hubig on 2019-02-15 [id 439]
(in reply to Report 2 on 2019-02-02)We would like to thank the referee for their reading of the manuscript and helpful comments and have replied in detail in-line.
We apologise for the oversight and have of course included a reference to this previous work on one-dimensional disordered systems.
We agree that it would of course be very nice to go to longer time scales, which we hope to firstly obtain by also considering higher-spin ancillas in future work as well as ongoing improvements to the numerical details of the algorithm. For the time being, however, we are limited in the time scale. This is not only due to the (remaining) growth of entanglement in the physical system but apparently also due to effective entanglement in the auxiliary system representing the many different disorder realisations of which we take the average. This second source of entanglement is not yet perfectly understood by us.
Comparing to other finite-size methods based on numerical linked cluster expansions, it appears that the referenced works find finite-size effects also at relatively short times (albeit in slightly different models). While a NLCE study of the system is not the subject of the current work, we might hence at least speculate that, applied to the current Hamiltonian, the method would likewise observe finite-size effects at times t < 1, well before our iPEPS method exceeds our computational resources.
We have reworked Fig. 3 and 4 to remove the unnecessary dashes and also reorganise the key in Fig. 3. We hope that the new figures are more readable.

---

## Round 1 · Referee Report · Anonymous · 2019-2-2

Strengths
1) first numerical study of iPEPS real-time dynamics with binary disorder average
Weaknesses
see report below
Report
The present manuscript studies the real-time evolution of a 2D S=1/2 Neel product state
under the action of the Heisenberg Hamiltonian with additional binary onsite magnetic field
disorder.
This is achieved using the iPEPS tensor network framework including an ancilla-trick to implement
the binary disorder average in a translationally invariant way. In 1D a (non-cited) work using iMPS
https://journals.aps.org/prl/abstract/10.1103/PhysRevLett.113.217201 performed an analogous study with
nice results.
The 2D case studied here is much more demanding, due to the challenging scaling of the computational effort with the bond dimension and some stability issues which are hard to cure because of the lacking
gauge fixing for iPEPS networks. Nevertheless the authors succeed to harness the stability to some extent and report simulations up to an impressive bond dimensions of D=9 (enabled by the Sz conservation).
The central results is likely Fig. 4, where the relaxation of the (staggered) magnetisation is tracked
as a function of time for various disorder strength. I have to admit that I am a bit disappointed that
the method is not able to access longer times in strong disorder case. I understand binary disorder is somewhat special, as the clusters of same-value disorder have finite probability even in the limit h-> infty, and the dynamics is not completely localised onto a single site, but still one might have expected
somewhat larger accessible time scales. Perhaps the authors could comment on this. Since the accessible
time scales are quite short, one is also wondering whether numerical or analytical short-time expansion approaches might be competitive. See e.g.:
https://arxiv.org/abs/1710.07696
https://journals.aps.org/prx/abstract/10.1103/PhysRevX.8.021070
https://journals.aps.org/prl/abstract/10.1103/PhysRevLett.120.070603
The figures are a bit hard to read for my taste due to the simultaneous change of
color, linestyles, symbols for different data sets. Furthermore the labels in Fig. 3
are clashing.
Requested changes
see report above.

---

## Round 2 · Author Response

Dear editor,
thank you very much for your work in organising and evaluating the
reviews of this manuscript. We have found the referee comments very
valuable and have improved the manuscript to incorporate those
comments. A brief summary of the bigger changes is listed below,
detailed replies to the referee's questions are posted on the version
1 submission page.
thank you very much for your work in organising and evaluating the
reviews of this manuscript. We have found the referee comments very
valuable and have improved the manuscript to incorporate those
comments. A brief summary of the bigger changes is listed below,
detailed replies to the referee's questions are posted on the version
1 submission page.

---

## Round 2 · List of Changes

- Incorporated a brief comparison to published 2D-NLCE calculations
- Extended the Conclusions to provide a wider outlook
- Clarified captions and improved legibility of figures

---

## Editorial Decision

published